# Too Low Motivation, Too High Authority? Digital Media Support for Co-Curation in Local Cultural Heritage Communities

**Edith Blaschitz** *,† , **Eva Mayr** † and **Stefan Oppl** †

Faculty of Education, University for Continuing Education, Arts & Architecture, 3500 Krems, Austria;
eva.mayr@donau-uni.ac.at (E.M.); stefan.oppl@donau-uni.ac.at (S.O.)
* Correspondence: edith.blaschitz@donau-uni.ac.at
† The authors contributed equally to this work.

**Abstract:** Over the last decades, a shift towards participatory approaches could be observed in cultural heritage institutions. In co-curation processes, museums collaborate with public audiences to identify, select, prepare, and interpret cultural materials. This article focuses on the question how to engage and motivate local communities or individuals in rethinking dominant discourses or expert narratives regarding cultural heritage and bringing in their own experiences and knowledge. Based on four case studies of cultural co-curation, we delineate two basic challenges for this process: (1) Authority—even though museums strive to involve the public, there is still an imbalance in participation due to the museums' authoritative status. (2) Motivation—participation in co-curation processes requires high levels of motivation, which are difficult to achieve. Based on the media synchronicity theory, we discuss which characteristics of new media technologies can be helpful to overcome these challenges. Media can increase awareness on counternarratives and blind spots in cultural collections. They can provide a setting where the participants can easily contribute, feel competent to do so, are empowered to rethink dominant discourses, develop a sense of relatedness with other contributors, and maintain autonomy in how and to which degree they engage in the discourse.

**Keywords:** co-curation; participation; cultural heritage; motivation; authority; digital media support; media synchronicity; local communities

## 1. Co-Curation for Cultural Heritage

Civic participation has a long tradition in museums and collections in order to engage and 'cultivate' the public audiences. Citizens even founded museums themselves as an expression of bourgeois emancipation in the 19th century. Progressive education movements in the 1920s and social reforms in the 1970s led also to considerations of how to involve audiences in central tasks of the museum such as collecting, preserving, researching, exhibiting and mediating narratives, practices and artifacts that are valued as cultural heritage [1]. A shift in society as a whole towards democratic participation set off again a "participatory turn" in museums at the beginning of the 2010s [2]. Participatory activities aim not only to achieve a higher level of audience engagement with collections or exhibitions, but also to promote previously underrepresented narratives or even "counter histories". In implementation, however, such activities are mostly restricted to crowdsourcing activities (e.g., collecting and commenting objects or crowd tagging). More recent participatory approaches also include collaborative exhibition design practices (in virtual and real spaces, cp. [3] and (p. 160, [4])), where collaboration and communication on archival objects aim to safeguard cultural heritage; In her highly influential work, Simon [5] describes four models of public participation in museums that characterize different forms of participation: *contributory projects*, where the audience has a small contribution in an institutionally controlled process, *collaborative projects*, where the audience becomes a partner in an institutionally controlled process, *co-creative projects*, where audience and institution

jointly control a process, and *hosted projects*, where the audience is in full control within the context of the institution.

This article focuses on the question, how to engage and motivate local communities or individuals to rethink dominant discourses or expert narratives regarding cultural heritage and bring in their own experiences and knowledge. Therefore, we will focus on *collaborative* and *co-creative projects* in this article, which allow and even require shared decision-making. Among the various definitions of collaborative processes, which are not always clear-cut, we have chosen the term co-curation, which Brewis et al. [6] (p. 4) describe as "*the negotiated identification, selection, preparation, and interpretation of materials.*" Crucial to this is a change in the role of the museum curator: In co-curation processes, which enable equal collaboration, curators switch to a "gatekeeper role" [7].

The advance of media technologies and of digital collections in the last decades give rise to new forms of **media-supported co-curation** which evolves within a complex socio-technical assemblage of curators, users, objects and machines. Media-supported co-curation "is performed in and through human and technical objects, relations and interactions, which are active in and simultaneously become organized through these platforms or the systems in which they operate" [8] (p. 11). In this sense, the design of media technologies becomes a decisive part of the co-curation process, which has the potential not only to support traditional processes of co-curation, but also to go beyond them and to overcome existing barriers for co-curation.

In this article we review several case studies of co-curation of cultural heritage and analyze existing obstacles to and challenges within these participatory processes. In particular, we explore the questions of how media technologies (1) can contribute to motivate people to engage in a co-curation process, in which artifacts, practices and narratives are collected, contextualized and interpreted in a collective way, and (2) can change the authoritative role of museums within this process. In this way, we capture the dichotomous nature of technology in co-curation, which can both, help to mitigate existing obstacles, but also induce new hurdles for participation if the socio-technical assemblage is designed in a way that does not account for the needs and prior experiences of the involved actors.

This article is structured as follows: in the next section, we present the conceptual foundations that serve as lenses to explore the role of media-support in co-curation processes. We then present four case studies that highlight different forms of co-curation processes and the challenges that can arise through digital media support or the lack thereof. In the following sections we synthesize the findings from the case studies along our research questions and elaborate on issues in and potential support for co-curation processes based the theoretical frameworks presented earlier. We conclude with a discussion of the overall findings and give some perspectives on the potential further development of the field and directions of future research.

## 2. Conceptual Foundations

Before presenting the case studies, we here introduce the conceptual lens that are adopted to explore the role of digital media support in co-curation in a structured way. We initially present Media-Synchronicity Theory [9] as an approach to explore the properties of digital media with respect to their support of different collaborative activities that are part of co-curation processes. As co-curation evolves in a socio-technical setting, which is not determined by technology alone, we in addition introduce Self-Determination Theory [10] as a foundation for explaining the motivational aspects of the social actors an their potential needs during co-curation.

### 2.1. Media-Synchronicity Theory

Co-curation activities are inherently collaborative activities. This collaboration, however, takes different forms for the different activities that are part of a co-curation process. From a technical perspective, these different forms also require different capabilities of the deployed digital media that are used to support the respective co-curation activity.

From a socio-technical perspective, each activity has to be supported by media with appropriate properties to enable task completion and also to lead to the desired basic motivational needs satisfaction. A mismatch between media properties and the features required to appropriately support the activities at hand lead to inefficiencies and eventually to frustration [11]—in particular in settings, where collaboration is supported by digital technology [12].

One approach to systematically assess different digital media regarding their appropriateness to support given collaborative tasks and to make informed selections is the Media Synchronicity Theory [9], which has proven a useful conceptual lens and framework for empirical studies on communication effectiveness and satisfaction in different contexts (e.g., [13–15]) and has also been shown to be applicable to examine motivational effects in online collaboration settings (e.g., [16,17]).

Media Synchronicity Theory (MST) distinguishes among two fundamental types of communication processes: *Conveyance* processes focus on sharing novel information with the collaborators in a way that allows them to individually understand them and integrate them with their existing knowledge. *Convergence* processes focus on developing a shared understanding about a topic of discourse among the collaborators, thus verifying, adjusting and negotiating individual understandings. A similar distinction between communication processes in the context of cultural heritage has also been proposed by Goodale et al. [18], who described the interplay between information access and sensemaking [19], but have not focused on the different affordances on media support in a systematic way.

Conveyance and convergence processes are interleaved in most collaborative settings, with a clear focus on one of them depending on the task (and the familiarity among team members, which can be put aside for this article, as scenarios of co-curation usually do not involve stable groups of people). Digital media selection thus must take into consideration the prevalent communication process and provide appropriate support.

In an attempt to systematically characterize media capabilities, MST describes five properties of digital media that impact whether they are more suitable for conveyance or for convergence processes. As visualized in Figure 1 and elaborated on further below, two of them, *transmission velocity* and *parallelism* are related to the transmission of information, whereas another two, *rehearsability* and *reprocessability* are related to information processing on the sender's and the receiver's side, respectively. The fifth property, *symbol sets*, is related to both: transmission and processing of information. In combination, all five properties contribute to or reduce the *synchronicity* of a given medium. In general, higher synchronicity makes a medium more suitable for convergence processes, whereas lower synchronicity indicates a fit to conveyance processes. This fit is further influenced by individual appropriation factors, such as familiarity, training, past experiences, or social norms. However, a fundamental misfit between the processes to be supported and the capabilities of the deployed media cannot be compensated for by appropriation factors.

Media that enable high synchronicity in the coordinated behavior of the participants usually have high transmission velocities (i.e., enable communication without delay in near real time) and offer a variety of symbol sets for communication (i.e., do not constrain information transmission on a single channel, but combine multiple modalities, e.g., audio, video, and text in online video conferences with screen sharing). Such media, however, will usually not allow for high parallelism (i.e., multiple participants contributing at the same time while still being able to distinguish individual contributions, as, e.g., would be possible in a forum or a shared document, but not in a video conference). Furthermore, high synchronicity usually reduces rehearsability (i.e., the option to check and possibly alter information before it is transmitted to the other collaborators) and also reprocessability (i.e, the option to repeatedly access information that was received at a former point in time). Conversely, media with lower synchronicity usually offer the option to rehearse and reproduce information before and after transmission, respectively. They also offer higher parallelism, enabling to convey a larger and more diverse set of information at a time.

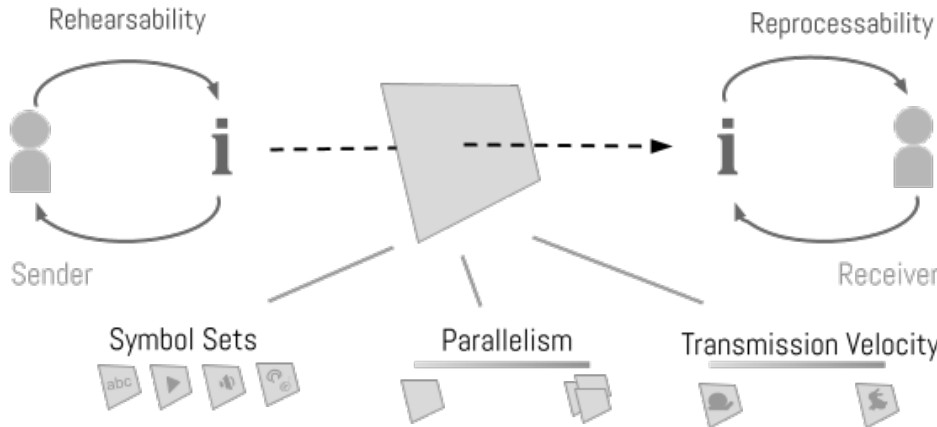

**Figure 1.** Media capabilities (adapted from [9]).

*2.2. Self-Determination Theory*

Motivational aspects play a significant role in the willingness to participate in co-curation activities. The antecedents of participants' individual motivation to engage in such co-curation processes can be explained by basic psychological need satisfaction and frustration as described and explored in Deci and Ryan's [10,20] Self-Determination Theory (SDT). The highest level of self-determined behavior is attained, if people act due to individual interest or fun (i.e., are intrinsically motivated) and thus experience inherent satisfaction. Still, motivation can be maintained also in other cases, if the values underlying the intended behavior are internalized. This internalization is bound to the degree of satisfaction of three universal psychological needs, that are essential "across individual and cultural differences" [21]: autonomy, competence, and relatedness.

The need for *competence* does not refer to "an attained skill or capability, but rather is a felt sense of confidence and effectance in action" [20]. People need to feel that they can successfully accomplish their tasks [10]. This can be facilitated through "well-structured environments that afford optimal challenges, positive feedback, and opportunities for growth." [10]. The need for *autonomy* refers to "self-governance" and does not imply "that people's behavior is determined independently of influences from the social environment" [10]. Providing options for individual choices can facilitate the sense of *autonomy*. Choices, however, only have positive effects when they are perceived to be personally significant [22]. *Relatedness* represents the desire to be supported, accepted, and to be a part of a community with shared interests and values [20]. One reason for acting in a motivated way is feeling related to other people that value the activity and convey "respect and caring" [10].

## 3. Case Studies

Having now explored the theoretical foundations that constitute the socio-technical assemblage in which co-curation processes evolve, we here now shift our focus to the practical aspects of differnet forms of co-curation. How can participatory processes of co-curation be built up? And what are common problems and challenges that arise? In the following, we present four projects, in which the potentials of co-curation have been explored for the identification and selection of objects, for the preparation of cultural content and for the interpretation of cultural heritage. The four projects and their evaluation were carried out at the Department of Arts and Cultural Studies at the University of Continuing Education Krems between 2016 and 2021, three of them with the participation of the authors. Digital technologies were used in different settings with the aim of engaging local communities. These are a biographical web platform that accompanied an exhibition, a mobile museum guide used to gain local knowledge about museum objects, digital

collection visualizations that allow users to explore the cultural material in their own way and an augmented reality app that makes historical events visible in public space. In all projects, the participation of local communities was a central goal, especially in order to generate expanded knowledge about the people, objects, or historical events being addressed. The projects serve as representatives of different types and phases of co-curation processes (as outlined in the definition given by [6] (p. 4)—identification, selection, preparation, interpretation) and thus have been selected for their diversity in the subjects of co-curation, the involved actors, and the focused-on co-curation activities.

The use of technologies was evaluated after the completion of the projects by the different project teams based on qualitative observations. Therefore and due to the pilot nature of the studies, no concrete numbers are available. The common challenges identified were not initially the focus of the projects and pilot studies, but are the result of the analysis and evaluation of the project outcomes. In reviewing them for common themes that arise in terms of encountered challenges with or without technology use, we set the stage for our further exploration of the potential roles of media technology in supporting co-curation processes.

### 3.1. Identification: Co-Curating an Exhibition and a Web Project

Historical women are hardly present in the public memory of Krems, Austria (as in many other cities), which means that there are hardly any streets named after women, no memorial for a named woman, and only few women mentioned in local history books. Therefore, in 2018 the participatory web project "*Memorial! Memorable?! Krems residents in search of their memorable women*" (Original German title: "DenkMAL! DenkWÜRDIG?! KremserInnen auf der Suche nach ihren denkwürdigen Frauen" https://raumforscherinnen.at/ (accessed on 28 January 2022, only in German language), project management: Martina Scherz, scientific support: Edith Blaschitz) invited residents to name memorable historical women with a relationship to Krems, who would deserve to be anchored in local memory. The call was distributed via the city's "official" offline and online channels (newspaper, digital newsletter, social media). The response was low, only about 20 references to historical women reached the curators, who added most of them to the website. The web-project was followed by the exhibition "*Where have they gone? The women of Krems*" (Original German title: Wo sind sie geblieben? Die Frauen von Krems, April–November 2021, https://www.museumkrems.at/Ausstellungen.htm (accessed on 28 January 2022)), conception and scientific coordination by Edith Blaschitz, Martina Scherz, curatorial team: Gregor Kremser, Sabine Laz, Doris Zichtl), which presented mostly unknown local female artists, scientists, politicians, teachers, journalists, entrepreneurs, activists or women in anti-fascist resistance in the Krems city museum "museumkrems" in 2021. Of these, 15 women came from local suggestions, 100 were additionally researched by the curators. In the exhibition again the audience was asked to name historical women and to pin the names on a wall with cards (see Figure 2 right). The amount of suggestions was somewhat higher, though there were a lot of affective contributions honoring grandmothers, mothers or friends.

**Problems, Challenges:**

- Although communicated via many different channels, participation in naming historical women was rather low, especially media-supported participation: An online form was provided on the website, but was not used at all, contact was made by phone or e-mail.
- A bias in the definition of "memorable" was observed: While the curators tended to use formal criteria such as educational qualifications, verifiable academic, artistic or political achievements, audience nominations were often women from their familiar living environment.
- Also a generational bias was observed: A younger audience did not actively participate in the co-curation processes online or in the exhibition, but rather only left "funny comments" in the exhibition.

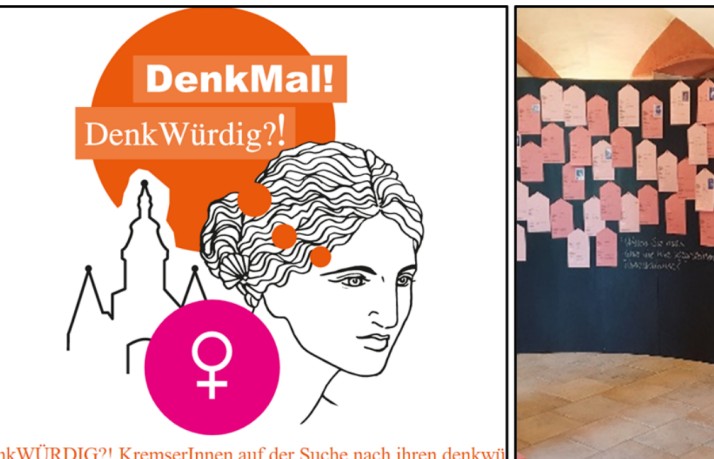

**Figure 2.** Web-project (**left**) and exhibition (**right**) on unknown and forgotten women in Krems.

### 3.2. Preparation: Co-Curating a Mobile Museum Guide

The mobile application Museumsmenschen (literally translated MuseumPeople, https://museumsmenschen.noemuseen.at/ (accessed on 28 January 2022) (only in German language); project lead: Anja Grebe, app implementation: Martin Reitschmied, Department for Arts and Cultural Studies, University for Continuing Education Krems) was developed for visitors of different city museums in Austria. With this app, visitors are involved in a dialogue with the museum founders on the museum, local history, and exhibits. In one of the museums, a participatory pilot project aimed to develop novel content in a co-curation process that involves not only museum staff, but also older citizens with their experiential knowledge of local history [23]. To overcome the challenge that older citizens have rich knowledge about the artifacts, but low technological literacy, the project team decided to establish mixed teams of teenagers and older citizens, based on the hypothesis that younger people are equally interested in cultural heritage and could contribute the technical skills necessary to create digital representations of artifacts. These intergenerational dyads with complementary knowledge on technology and on local history successfully collaborated on novel artifact stories for the mobile application.

**Problems, Challenges:**

- Older citizens have high knowledge on local history and can make meaningful contributions in co-curation processes, but frequently also have lower technological literacy, which poses a barrier for the participation in technology-mediated co-curation.
- Even though they are similarly interested in cultural heritage than other age groups, it turned out to be difficult to find teenagers who volunteered to contribute in a co-curation process on cultural heritage topics in their leisure time (though those who participated evaluated the project quite positively). In the end, the participants were recruited based on direct personal contact, an open call on the web and in local newspapers worked for the older citizens, but not for the teenagers.

### 3.3. Interpretation: Co-Curating with Collection Visualizations

In the last decades many cultural collections have been digitized and made accessible as digital collections on the web. Though these museum databases give access to a rich collection of objects, they are not easily accessible by non-expert users [24]. Collection visualizations offer an alternative, more generous way to explore these databases and to support the interpretation of cultural heritage [25]. Visualizations combine the data processing powers of computers with the perceptual and cognitive powers of humans. In the case of cultural collections, they provide an overview on the digital collection to support users' exploration as well as the identification of patterns and objects of interest for close viewing. In several research and development projects over the past years, the second author collaboratively developed interactive visualizations which enable and support

the open-ended, serendipitous exploration of cultural collections (see Figure 3 for two examples: The visualizations have been developed for two different Austrian collections, the left visualization can be explored at https://visualisierung.landessammlungen-noe.at/ (accessed on 28 January 2022), the right one is in a prototype stadium). These visualizations allow users to explore the cultural material in their own way and see multiple facets of the cultural collection.

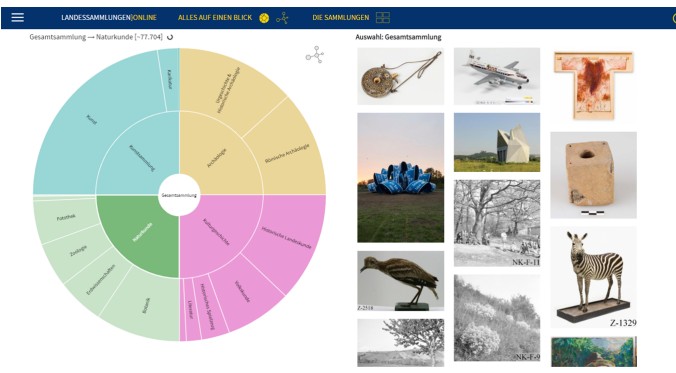 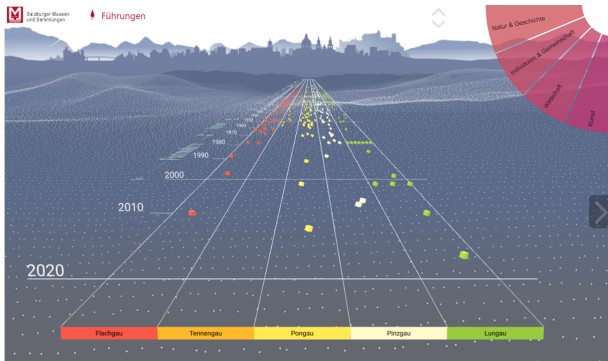

**Figure 3.** Collection visualizations based on a hierarchical classification of objects (**left**) and their geo-temporal origin (**right**).

**Problems, Challenges:**

- Though serendipity is a declared a target aim of collection visualizations [26], it remains an open question, whether they actually can motivate users "in the (world) wild (web)" to engage with the collection over a period long enough for deeper processes of meaning making and interpretation.
- In this asynchronous setting, where the curators and the visualization developers provide a setting for interpretation by users later on, no control on the users' interpretation can be exerted. Still, the selection of relevant data dimensions within
- A further challenge would be to develop novel collaborative visualization approaches, which allow visitors to share their interpretation with other users or to let others develop their interpretation further.

*3.4. Interpretation: Co-Curating Memory Processes in Public Space*

The augmented reality app "Zeitreise Tabakfabrik" (*Time Travel Tobacco Factory*) (Conception and scientific coordination by Edith Blaschitz, implementation: Center for Applied Game Studies, Department for Arts and Cultural Studies, University for Continuing Education Krems, https://www.donau-uni.ac.at/de/universitaet/fakultaeten/bildung-kunst-architektur/departments/kunst-kulturwissenschaften/zentren/stabsbereich-digital-memory-studies/forschung/projekte/zeitreise-tabakfabrik.html (accessed on 28 January 2022)) tells the history of the Stein tobacco factory, using film sequences taken from a historical documentary produced in 1927 about the everyday working life of female tobacco workers. The factory, opened in 1850, was closed in 1991 and the factory building has been transformed into a university, with a corresponding infrastructure consisting of offices and teaching rooms. Via smartphone or tablet, the digitized historical film clips of female workers can be viewed at the actual site of the event (see Figure 4).

During guided tours, which have been taking place since 2015, not only the augmented reality app can be used, but also former workers participate who report on their prior working life. Thus the visual perception of images at the site of the event are complemented by oral narratives. These two components together often trigger spontaneously new narratives and memories. Thereby, new knowledge emerges through interactions between digital technologies, eyewitnesses, historical sites and the audience, who often has a specific relationship with the factory or the workers. In a second project phase the audience was

asked to recount memories on the tobacco factory or workers. These stories were video recorded and are to be integrated into the updated app.

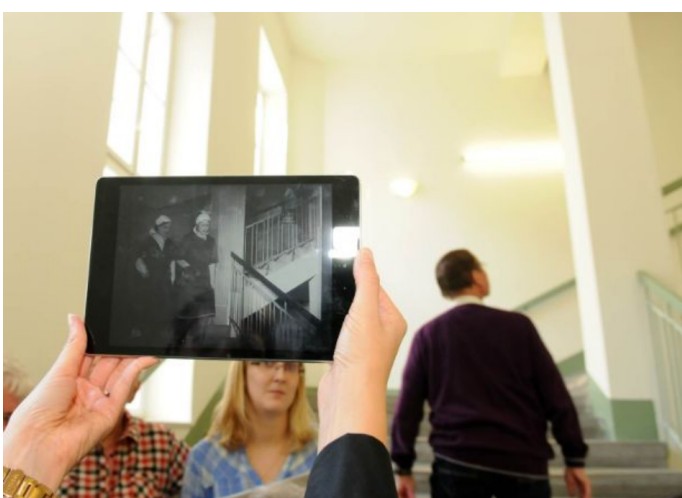

**Figure 4.** Guided tour with the augmented reality app "time travel tobacco factory".

**Problems, Challenges:**

- Older participants did not want to use the tablets provided to access the videos via the app. Younger ones took over the handling of the tablets, acting as intermediaries.
- The shown visual narratives gave the impression of seeing the authentic past—"how it really was". It was only when the eyewitnesses came up with different stories and perspectives that the audience was stimulated to talk about deviating experiences and memories.

*3.5. Common Challenges*

Even though all four case studies address different aspects and phases of co-curation, we observed common challenges in all of them regarding (1) authority and (2) motivation.

The first set of challenges, which emerged in most co-curation settings, are related to **authority**: It became evident in the first case study that the museum's criteria for characterizing a woman as "memorable" did not always correspond to those of the audience and the users. In the second case study on the preparation of cultural material for a multimedia guide, elder citizens were attributed higher authority to contribute relevant experiential knowledge in contrast to younger ones. The collection visualizations in the third case study appear to be open for self-directed exploration and interpretation of the collection by interested users, but the compilation of objects within the collection and the selection of relevant metadata dimensions for the visualizations restrict these perceived degrees of freedom by authoritative decisions taken by the museum. Finally, in the last case study the visitors took the authority of the media material for granted based on the authority of the authoring institution. Only those visitors with contrasting experiential knowledge did question their authority based on inconsistencies between the material and their experiences.

The second set of common challenges relates to the question of **motivation**: In several case studies it was difficult to motivate people to participate in open calls on the web and in newspapers. Especially younger groups cannot be easily activated, which might be also related to the question of authority due to a perceived lack of relevant experiential or cultural knowledge. In uncontrolled co-curation settings on the web (like in the third case study) the motivation to participate cannot be controlled at all—despite via the design of the technologies. In two case studies we observed that media technologies can even be part of the motivational obstacle, as some (especially elder) target groups miss the technological self-efficacy and skills necessary to contribute to media-supported co-curation processes.

In the following chapters, we elaborate on these two sets of challenges and examine from a theoretical perspective, how digital media could contribute to address these challenges in the process of co-curation in cultural heritage. We are aware that these challenges can and should also be addressed with non-media measures, but these are beyond the focus of this article—here we explicitly focus on how digital media influences co-curation processes and can help overcome the identified challenges.

## 4. Questioning Authority in Digital Media for Co-Curation

Why is authority such a challenge in co-curation processes? When participants are invited to interact with the museum's collections, the museum often retains the interpretive authority over the definition of cultural heritage, as artifacts kept in museums as cultural heritage have gone through a process of evaluation and classification. Mostly, museum professionals decide on the 'museum-worthiness' of objects. Collecting and curating are processes strongly influenced by dominant narratives and discourses (see Figure 5). If artifacts, social practices and narratives do not conform to the prevailing social interpretation of cultural heritage they are often not valued as such and not included in museum collections [27] or objects brought in are subordinated to the 'expert' narratives produced by the official institutions [28]. These narratives may differ from the experiential knowledge communities possess [29]. In contrast, local communities or individuals may consider entirely different artifacts, practices and narratives as their cultural heritage worth preserving and passing on to future generations, which do not follow the dominant narratives and discourses. Following a cultural studies-oriented conceptualization of "culture" the interpretation of cultural heritage in local communities is anchored in the everyday life of community members ("a whole way of life", [30]). In this sense, cultural heritage generates its meaning through contextualization in personal narratives and social interactions of people that directly or indirectly (e.g., via ancestors' stories) can relate to the tangible or intangible cultural heritage of interest.

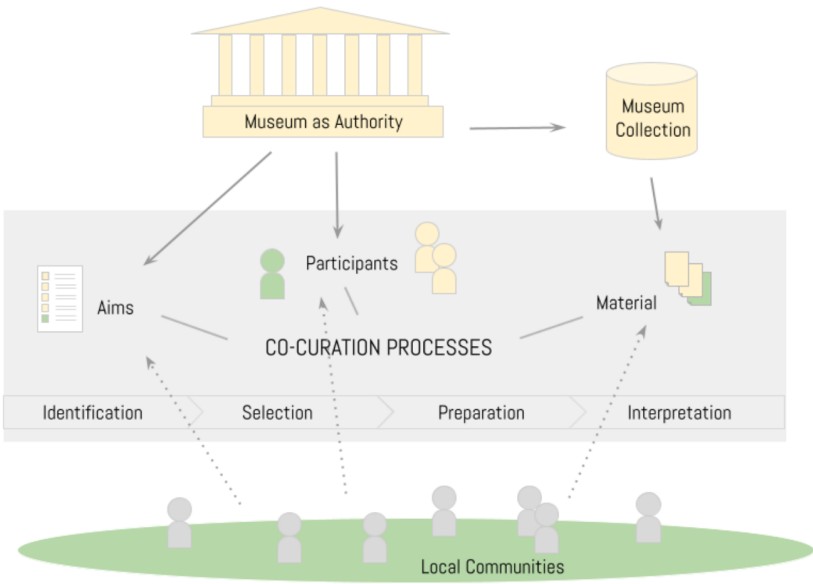

**Figure 5.** Imbalances in co-curation processes.

Digitization measures can increase the bias by perpetuating and reinforcing dominant metadata schemes and tagging (Aghostino 2019), but media-supported co-curation has also the potential to reduce existing imbalances of authority and novel media interfaces can "challenge long-standing assumptions and biases embedded in cultural collections" [31] (p. 114). To reach this potential, we have to design (technology-mediated) co-curation processes which trigger critical, reflective processes, so that the participants do not take the material or information as granted based on some attributed authority.

Critical thinking is an important skill frequently studied in (computer-supported) learning (e.g., [32,33]) and workplace settings (e.g., [34,35]). In informal settings like museums critical thinking was only seldom addressed and if so mostly on an individual level [36,37], but not with respect to co-curation processes. It is important to be aware that processes of reflective, critical thinking do not occur automatically (as taking facts as granted requires much less mental effort), but are triggered by discrepancies which attract our attention: Contradicting information, mismatches with our prior knowledge, (social) conflicts, or highly interesting novel information are just some examples.

Media technologies can support the awareness of multiple perspectives within co-curation processes. In an experimental study, it could be shown that the saliency of different positions in an opinion terminal did enhance critical thinking in museums [37]. It is therefore an important role of media technologies to increase the awareness on multiple narratives, viewpoints, and positions in parallel and thereby trigger reflective processes. In a similar fashion, Glinka, Meier and Dörk [31] propose to develop critical visualizations which show a cultural collection as a whole on one glance and allow users to identify master narratives as well as "blind spots" (i.e., geographical regions or time periods without any or with only a small number of objects), so that users can contribute items to exactly these spots and can help to fill them up. However, we need to "assure that this new layer of information supports diversity and does not only present a shift from one exclusionary process to another" [31] (p. 112).

Next to making participants aware of multiple perspectives and imbalances, investing mental effort in (cognitively demanding) processes of critical thinking also requires motivation, which also can be supported by media technologies.

## 5. A Motivational Lens on Media Support for Co-Curation

As could be observed in all four case studies, motivational aspects played a significant role in the willingness to participate in media-supported co-curation activities.

Using the lens of Self-Determiation Theory as an explanatory framework, this means that participants must be provided with a (socio-technical) setting where they can contribute, feel competent to do so, are empowered to rethink dominant discourses, develop a sense of relatedness with other contributors, and maintain autonomy in how and to which degree they engage in the discourse. Media technology can play a dichotomous role here—while its informed deployment can contribute to satisfy these basic psychological needs, it can also induce new hurdles for participation in particular for target groups that are not accustomed to used digital technology in their everyday lives (e.g., as shown in [38]).

Empirical evidence (e.g., [39]) shows that people engage in technology-mediated interaction based on their individual prerequisites (e.g., such as prior experiences), and contextual factors in these settings can contribute to improve the quality of their motivation to engage with digital technology. Rather than feeling pressurized to achieve externally imposed demands, fostering motivational processes of internalization and integration can help individuals to overcome the lack of feeling competent and to realize the potential of using digital technology for their individual aims.

In the different stages of co-curation processes (cf. [6] (p. 4)), the contextual factors that influence the satisfaction of needs to be considered vary, as could also be observed in the case studies. Therefore, when designing media support for co-curation, different basic psychological needs should be satisfied.

For *identification* processes, people have to be provided with easy, non-blocking gateways to provide input to the co-curation process. This requires adequate media support to enable low-threshold, self-directed capturing of materials in different modalities, such as text, images, audio and video without challenging participation by imposing technical constraints or setting artificial organizational or technical hurdles such as registration forms, logins or the need to explicitly request access.

Similarly, engaging in *selection* processes in an autonomous way may be interrupted by technical constraints that pose ex-ante challenges in participation. In contrast to the identification phase, selection activities build upon already existing materials and thus require digital media support to provide structured access to these materials and appropriate means to contribute to selection in a collaborative way (e.g., voting systems or opportunities for qualitative feedback and discussion).

*Preparation* activities in co-curation add to the complexity of selection activities the need to support modifications of existing materials, contextualizing them, and putting them into mutual relationships with each other. In terms of media use, this phase puts high demands on technology support as participants have to act autonomously (even more so if preparation is carried out collaboratively) and feel competent not only in the domain of curation but also in the media use itself. Here, the common trade-off between restricting freedom of choice to strengthen ease of use and striving for openness in how people can interact with materials and each other via interactive technology becomes obvious [40].

Finally, *interpretation* activities are similar to identification activities from a technology-centric point of view with respect to their potential impact on motivation. Capturing of interpretative additions to materials requires low threshold gateways for media creation to avoid technology-induced frustration and lacking sense of competence. One specific challenge of this phase is the need to make visible the "who" in interpretational materials, as perspectives here explicitly need to complement each other. Providing ways of specifying one's perspective and at the same time maintaining control about the degree of which one's own "real-world" identity is made transparent is a challenge that also needs be addressed via technological means and—if available—can contribute to one's sense of autonomy (in terms of "feeling free" to speak without fear of non-acceptance or even retribution).

In general, observations from the case studies indicate that deficiencies in satisfying the need for competence and autonomy can at least partially be compensated by strengthening relatedness, e.g., through collaborative work on tasks (as shown in case study 2), or support through scaffolding activities (i.e., providing situated and need-driven support, e.g., as described by [41]).

## 6. Digital Media for Co-Curation—An Account on Technical Design Dimensions

In the context of co-curating cultural heritage in an open, participatory way, Media-Synchronicity Theory can be used to reflect on the different activities that take place along the process. We here again follow the activities referred to by Brewis et al. [6]. It is important to note here that not one single media configuration will likely be appropriate for all activities. Rather, supporting media must be selected according to the respective activity and be tailored to the specific situation at hand. MST here acts as a conceptual lens to structure the process of analyzing a specific co-curation scenario and of selecting appropriate media in an informed way. In the following, we thus outline generic considerations for media selection along the different co-curation activities, which need to be contextualized in the scenario at hand to be put to practice.

The nature of *identification* activities as characterized in the case studies indicates that conveyance processes are the prevalent mode of collaboration here. This calls for media with low synchronicity, where people can reflect on and potentially alter their contributions before submitting them. At the same time, high parallelism is required to enable broad participation without the need to coordinate individual contributions upfront.

*Selection* activities require a different mode of collaboration, as different perspectives and counternarratives need to be explicitly made visible, if this process is to be organized in a collaborative way. This calls for a focus on convergence of understanding on which individual perspectives are present to eventually also identify potential blind spots. Such activities likely would require higher synchronicity and more immediate exchange of perspectives on different materials. These perspectives not necessarily have to be accessible for future reference, but are only relevant in-situ, until a deliberate decision has been reached among the contributors (i.e., reprocessability is not relevant here).

While the initial two sets of activities focus on the intake of cultural heritage for further processing, preparation and interpretation happen later on, when initial curation decisions have been made. *Preparation* is characterized by activities that focus on contextualizing materials and making them understandable to others. This requires deliberate consideration of which information to provide in which way, potentially with a variety of symbol sets. Such activities in turn call for media with low synchronicity, that enable rehearsability of information, before it is put out for others to access. Complementary, *interpretation* activities are only possible, if information on cultural heritage materials is reprocessable several times and thus can be reflected on, before it is interpreted. Interpretation from multiple perspectives also required high parallelism and should deliberately avoid to enforce the development of a single common understanding of a particular artifact of cultural heritage. As such, interpretation mainly builds upon conveyance processes and demands media with low synchronicity. Preparation activities are characterized by an interplay of conveyance and convergence processes, which are equally important here and thus set diverse requirements on media support.

It should be noted here that media synchronicity theory is not restricted to analyze digital media, but can be used to systematically assess the appropriateness of any collaborative setting, whether it be technically or socially mediated. In combination with our considerations on aspects of authority and motivation, a perspective on the capabilities of socio-technical media for collaboration offers valuable perspectives on the design of digital and physical spaces to enable and facilitate co-curation processes.

## 7. Discussion & Outlook

In early papers on media-supported co-curation, it was supposed that these participatory processes result "in richer discourse through globally dispersed public participation and intersubjective perspectives" [42] (p. 589). Experiences like those presented in the four case studies make clear that in order to motivate local communities or individuals to rethink dominant discourses or expert narratives regarding cultural heritage several obstacles have to be overcome before such a rich discourse can result. Among them, we regard the challenges of authority and motivation as the most prevalent ones. To overcome these barriers we suggest negotiated and collaborative co-curation processes, as we have described them above. Participants of such processes must be provided with a (socio-technical) setting where they are aware of multiple positions and narratives, are empowered to rethink dominant discourses, can contribute, feel competent to do so, develop a sense of relatedness with other contributors, and maintain autonomy in how and to which degree they engage in the discourse. On the other hand, such a collaborative process also counteracts a dominant attitude of museum experts, who often still have problems accepting community contributions that contradict their point of view. Such a setting requires informed selection of a set of digital media for appropriately supporting these collaborative tasks along the different phases of co-curation. Media Synchronicity Theory can be used as a framework for such informed decisions, as it allows to explicitly consider the nature of a particular communication setting with respect to the desired outcomes (conveying different perspectives vs. converging on a common understanding along the different activities of co-curation). These communication towards these different outcomes are facilitated or hampered by particular capabilities of different media. Explicitly accounting for the fit between required support and actual capabilities of media can help avoid design decisions that lead to technology being perceived as a hurdle rather than as an enabler and facilitator.

Media-supported co-curation processes involve a socio-technological system of heterogeneous participants (with more or less expertise on different aspects) and a set of digital media, that mediate the processes in different ways by creating awareness on existing imbalances, providing material, offering a communication platform or a group thinking tool. Though digital media can solve some problems, we agree with Mutibwa et al. [4] (p. 166) that they are not the solution to all existing challenges, they even pose novel ones, as some target audiences reject digital media. But bottom-up initiatives such as the

web-based Austrian cultural collaboration platform Topothek [43], in which independent local units collect, digitize and contextualize historical images and artifacts, prove that digital environments are also well received at a local level and can make local cultural heritage more visible.

On the social side of the media-supported co-curation process, it should be carefully considered who are the relevant target groups for cultural co-curation. Established communities of interested museum visitors, for example, are often already intrinsically motivated (due to inherent interest and fun in the topics at hand), can be more easily reached, and come with higher levels of prior knowledge (e.g., [4]) than unrelated crowds, which might benefit more from additional media support and/or stimuli to participate in a motivated way. The attachment to formal institutions such as the museum can be the stimulus for further engagement in media-supported environments. On the other hand, these frequent museum visitors are maybe also more compliant with the authoritative mainstream knowledge of the museum institution and less suited to bring in more critical counter-positions and -narratives.

In this paper, we used the term digital media for a broad range of technologies, which can support co-curation: from collaboration platforms on the web, to cognitive tools which support co-curation processes and reasoning, to technologies for the communication of the outcomes of the participatory process. We want to encompass the whole heterogeneity as each one of them can support different aspects of co-curation and might be of use in different participatory projects. Despite their different forms and applications, we contend that digital media can support co-curation, but they have to be carefully designed to overcome challenges of authority and motivation inherent to participatory processes in cultural heritage and to mitigate challenges inherent to the technologies themselves.

**Author Contributions:** Conceptualization all authors, Sections 1, 3 and 4, E.B. and E.M., Sections 2, 5 and 6, S.O., Section 7 and review all authors. All authors have read and agreed to the published version of the manuscript.

**Funding:** Case Study 1 received funding from kremskultur, Bundeskanzleramt, Kultur Niederösterreich. Case Study 3 received funding from the state Salzburg.

**Institutional Review Board Statement:** Not applicable.

**Informed Consent Statement:** Not applicable.

**Data Availability Statement:** Not applicable.

**Conflicts of Interest:** The authors declare no conflict of interest.

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
