# Peer review of "Too Low Motivation, Too High Authority? Digital Media Support for Co-Curation in Local Cultural Heritage Communities"

_mti, doi:10.3390/mti6050033_

Round 1

Reviewer 1 Report

The presented contribution deals with the interesting issue of co-curation in local cultural heritage communities. The article provides 4 case studies identifying problems and challenges.
Line 137: Are there any numbers available? Any study to confirm or refute this claim?
Why are problems and challenges (without bibliographic or numerical support) generally identified in specific case studies?
Challenges are addressed only at the philosophical level. The identification of the problem is OK, the subsequent concrete solution, at least on a theoretical level, is absent in the article. For example, the older generation (generally focused on history) and the younger generation (focused more on technology) suggest specific possibilities for developing co-operation in co-curation. In what specific way can these different groups be motivated so that the result is as effective as possible and co-heating has a positive impact on all groups? From my own experience, I know that there are exceptions to the older generation (approximately 70 years) and individuals who are fascinated by modern technologies and their own motivation create opportunities to promote digital media for curation and co-curation in local cultural heritage communities. Similarly, the young generation is not losing interest in history. Is there any way to implement and take this into account in the article?

Author Response

Thank you very much for the positive evaluation of our paper and for the thoughtful, constructive feedback!

We adapted our paper as follows:

  • @Reviewer 1 / Line 137: Are there any numbers available? Any study to confirm or refute this claim? Why are problems and challenges (without bibliographic or numerical support) generally identified in specific case studies?
    As case study 2 is only a small pilot study, there are no concrete numbers on these aspects. The case-study specific problems and challenges are based on qualitative observations made by the different project teams and do not claim to be valid beyond them. In section 2.5 we generalize by comparing the observations made in the different case studies and support them with bibliographic references and theories in sections 3 and 4. 
  • @Reviewer 1: From my own experience, I know that there are exceptions to the older generation (approximately 70 years) and individuals who are fascinated by modern technologies and their own motivation create opportunities to promote digital media for curation and co-curation in local cultural heritage communities. Similarly, the young generation is not losing interest in history. Is there any way to implement and take this into account in the article?
    We have adressed this issue on a conceptual level in the first paragraph of section 4 when introducing SDT by explicitly stressing that acting due to intrinsic motivation does not depend on basic need satisfaction. Only if intrinsic motivation is not present, satisfying the needs for autonomy, competence and relatedness can support to still act in a motivated way. Additionally, we have also explicitly adressed this issue in discussion (section 6, paragraph 3), where we now delineate intrinsic motivation as not being dependent on additional media support in the context of the museum scenario.

For track changes, please visit the article on overleaf: https://www.overleaf.com/read/fyhqqdnthfkz

Reviewer 2 Report

An excellent study of the issues surrounding community engagement with museum collections. The focus on online methods is highly relevant at present

Very interesting discussion of the interplay of authority and motivation. Maybe one point missed in the authority discussion is the bias of museum staff who will not choose or hear community contributions that conflict with their  view of history, as community are only called in at key points.

Focus on encouraging reflection and critical thinking to bring in new perspectives. However as in the motivation study this is hypothetical

A good approach to look at the different factors in motivation, however the use of Media Synchronicity Theory abstracts maybe too far from the context under study, as the intake of cultural heritage material is blurred with the later curation by the museum.

It is a pity the details of the interface types for content input, rather that just the technology (ipad etc) was not included in all the case studies, what can be done in each system and what can people see when contributing 

This may also be a factor in the motivation and authority factors

Author Response

Thank you very much for the positive evaluation of our paper and for the thoughtful, constructive feedback!

We adapted our paper as follows:

  • @Reviewer 2: A good approach to look at the different factors in motivation, however the use of Media Synchronicity Theory abstracts maybe too far from the context under study, as the intake of cultural heritage material is blurred with the later curation by the museum.
    We have adressed the issue of granularity / blurring between phases in co-curation explicitly in section 5, paragraph 7 (just after the generic introduction to MST) and here now make more visible, that successful media selection is dependent on a detailled decomposition of the specific co-curation setting. In addition, this issue has also been picked up in the discussion section. 
  • @ Reviewer 2: Maybe one point missed in the authority discussion is the bias of museum staff who will not choose or hear community contributions that conflict with their  view of history, as community are only called in at key points.
    We have included the perspective of museum experts in the process of co-curation to show that it also helps to counteract the bias present here.

For track changes, please visit the article on overleaf: https://www.overleaf.com/read/fyhqqdnthfkz

Reviewer 3 Report

The paper's subject is interesting and very well presented. The case studies are successful examples related to the theme of the paper and the conclusions complete. The paper is supported by up to date references. I suggest to be published in its current form.

Author Response

Thank you very much for the positive evaluation of our paper!

Reviewer 4 Report

This paper presents a theoretical discussion on co-curation in cultural heritage.  Several case studies are reviewed and the obstacles and challenges in these participatory processes are analyzed. It highlights the dichotomous nature of technology in co-healing, which can help mitigate existing barriers, but also induce new barriers to participation if the socio-technical assembly is designed in a way that does not consider the previous needs and experiences of the stakeholders involved.

In my opinion, the article is well suited for the special issue to which it has been submitted.

The bibliography is comprehensive and up to date. The document includes an extensive work of synthesis and search for connections between previous works, presented in an adequate way. The opinions expressed by the authors are also well supported by the reasoning provided.

In summary, I think that it is a well-worked document that can be published in its current form.

Congrutalations on your work!

Author Response

Thank you very much for the positive evaluation of our paper and for the thoughtful, constructive feedback!

Round 2

Reviewer 1 Report

No comment. My questions were answered and the article adjusted.

Author Response

Thank you for your approval!